# Sources of Variability in the Response of Labeled Microspheres and B Cells during the Analysis by a Flow Cytometer

**DOI:** 10.3390/ijms22158256

**Published:** 2021-07-31

**Authors:** Adolfas K. Gaigalas, Yu-Zhong Zhang, Linhua Tian, Lili Wang

**Affiliations:** 1Fluorescence Consultant, Riviera Beach, FL 33404, USA; 2Thermo Fisher Scientific, Eugene, OR 97402, USA; Yu-Zhong.Zhang@thermofisher.com; 3Biosystems and Biomaterials Division, National Institute of Standards and Technology, Gaithersburg, MD 20899, USA; linhua.tian@nist.gov (L.T.); lili.wang@nist.gov (L.W.)

**Keywords:** stochastic model, flow cytometer, coefficient of variation, antibodies bound per cell

## Abstract

A stochastic model of the flow cytometer measurement process was developed to assess the nature of the observed coefficient of variation (CV%) of the mean fluorescence intensity (MFI) from a population of labeled microspheres (beads). Several sources of variability were considered: the total number of labels on a bead, the path through the laser beam, the optical absorption cross-section, the quantum yield, the numerical aperture of the collection optics, and the photoelectron conversion efficiency of the photomultiplier (PMT) cathode. The variation in the number of labels on a bead had the largest effect on the CV% of the MFI of the bead population. The variation in the path of the bead through the laser beam was minimized using flat-top lasers. The variability in the average optical properties of the labels was of minor importance for beads with sufficiently large number of labels. The application of the bead results to the measured CV% of labeled B cells indicated that the measured CV% was a reliable measure of the variability of antibodies bound per cell. With some modifications, the model can be extended to multicolor flow cytometers and to the study of CV% from cells with low fluorescence signal.

## 1. Introduction

This manuscript describes a stochastic model of a flow cytometer measurement of fluorescence emission from microspheres (beads) labeled with immobilized dye molecules. The model describes the dependence of the measured coefficient of variation, CV%, on instrument parameters, as well as the properties of the labels in a population of beads. The stochastic model complements a stochastic model of the labeling reaction, described in a previous paper [1]. The two models provide a basis for interpreting the coefficient of variation (CV%) of the mean fluorescence intensity (MFI) from a population of labeled beads analyzed by the flow cytometer. In most cases, the variation in the number of labels on the beads gave the largest contribution to the measured value of CV%. The stochastic model also provides estimates of the contribution to the CV% due to variability of the fluorescence detection alone and the variability of the background signal. Together, the two estimates provide a correction to the measured CV% and give a reliable estimate of the CV% of the number of labels on the bead. This work demonstrates that a similar correction can be made to measurements on cells and provide a reliable measure of the heterogeneity of the number of bound labeled antibodies on the cells.

The stochastic model, described below, consists of two major parts: first, the production of the fluorescence photon stream by the illuminating laser and microfluidics of the flow cytometer, and second, the detection of the fluorescence photon stream by flow cytometer collection optics and electronics. The model is first applied to a set of multilevel labeled beads. The analysis of labeled beads by a flow cytometer is well understood [2,3] and serves as a system to validate the stochastic model. The beads were modeled by randomly placing a number of labels inside or on the spherical surface of the bead. The results obtained for the bead system were applied to the estimate of the CV% of labeled mAb on the surface of B cells.

The number of fluorescence photons emitted during the passage of a bead through the focused laser beam of a flow cytometer depends on the number of labels on the bead (N_lb_). The fluorescence from any one label depends on the intensity of the incident laser beam (*I_o_*) at the location of the label, the optical absorption cross-section (σ), and the fluorescence quantum yield (QY). The optical absorption cross-section depends on the relative orientation of the transition dipole moment of the label molecule and the laser polarization. It is likely that the transition dipole orientation, when averaged over all labels, gives a constant average optical absorption cross-section for the beads in the sample. The QY depends on the label environment, but the average over all label environments on the bead gives a constant average QY for the beads in the sample. Therefore, the resulting stream of fluorescence photons is proportional to the product of *I_o_*, N_lb_, the average absorption cross-section σ, and the average quantum yield QY. It is assumed that the electric field profile at all points in the bead is the same as the electric field profile of the freely propagating focused laser beam. Distortion of the electric field profile by the water–bead interface is neglected in this version of the model. In Section 2, the model was used to calculate the response of beads transiting through the laser beam and estimated the resulting photon stream. The main sources of variability was the variation in the number of labels on the beads in the sample and the variation in the path of the bead through the laser beam. The fluctuations of the average optical absorption cross-section and QY were relatively small, and they are set to zero in the current version of the stochastic model. Section 2.2 details the collection of the emitted fluorescence photon stream by the detector entrance aperture and photoconversion at the photocathode of a photomultiplier (PMT). The resulting electron stream was converted into a voltage pulse, digitized, and displayed on the computer screen. The MFI was estimated by simulating the response of several hundred bead transits and calculating the mean of the maximum fluorescence signals of the transits. The CV% was obtained from the spread of the maximum fluorescence signals. Section 2.3 presents some measurements of beads labeled internally with FITC dye and applies the stochastic model to the analysis of MFI and CV% from the beads. An excellent introduction to the operation of a flow cytometer can be found in Chapter 2 of [4]. Other chapters provide a guide to the vast number of applications of flow cytometer measurements.

## 2. Results

### 2.1. Measurement of Multilevel Bead Response

The model was compared to bead measurements which have been studied extensively and serve as the best model system [5]. Figure 1 shows the results of Attune NxT flow cytometer (FC) analysis of a sample containing four latex bead populations, each with progressively larger loading of dye molecules (beads obtained from Thermo Fisher Scientific). The bead diameter was 5.2 µm. The horizontal axis, called BL1-H, shows the height of the fluorescence photon pulse detected by the BL1-H channel when a bead passes the 488 nm laser beam. The vertical axis shows the number of times a particular signal was observed during the FC analysis of the bead sample. In practice, about 10,000 beads were included in the analysis. The lowest peak, indicated by B in Figure 1, corresponds to the bead population without dye molecules and serves to estimate background signal due to autofluorescence and Raman scattering from the latex material and water, as well as feedthrough of elastically scattered light. The events in the four peaks were gated with a strong forward scattered signal detected by a photodiode such that the events are true fluorescence signals associated with a passage of the bead. The three stronger peaks in Figure 1, indicated by L, M, and H are due to beads containing different amount of dye molecules and were not present when only the unlabeled blank beads, B, were analyzed.

Each peak in Figure 1 can be assigned a mean fluorescence intensity (MFI) and a standard deviation (SD) of the spread of intensity values in the peak. A compilation of the measurements is given in Table 1 for a FC with a laser with a Gaussian beam profile and Table 2 for a FC with a laser with a flat-top laser beam profile. The measurements were used to estimate the contribution of the fluctuations in the measurement process to the measured CV% of the beads. Each table contains three sets of measurements, each with a different sample flow rate given by 100 µL/min, 200 µL/min, and 500 µL/min. As expected, the height of the peaks does not depend on the sample flow rate. The beads experience the same maximum illumination irrespective of how fast they transit the laser beam. Below, we use the data for a flow rate of 100 μL/min for further analysis.

The measured values presented in Table 2 were compared with the values obtained from the model, which provided detailed contributions to the CV% of the labels on the various beads. Model calculations (not shown) with a Gaussian profile laser beam gave a larger contribution to the detection CV% than the flat-top laser profile. Specifically, for the M bead, the CV% from detection variation was 0.35 when the path of each bead through the laser beam was fixed. The CV% increased to 0.95 when a variation of ±2 μm was applied to the *x*-coordinate of the path of each bead transiting the laser beam. In contrast, the detection variation for the flat-top laser resulted in a CV% of 0.22 for a constant path and 0.24 when the *x*-coordinate of the path varied by ±2 μm. Since we want the least CV% contribution from detection/instrument fluctuations, we chose the measurements with the flat-top laser for further analysis. The flow rate in Table 1 and Table 2 did not significantly affect the values of MFI and CV%. Figure 2 shows a typical time series of digital voltage values for a single transit of a bead. The blue dots and orange dots are for beads with a linear velocity of 1.6 m/s and 0.8 m/s, respectively. Note that the maximum peak height is independent of the linear velocity. Therefore, the model focuses on one linear velocity, 0.8 m/s. Voltage pulses were obtained for 248 bead transits. The average pulse height and the SD were obtained from the 248 values. The histogram made from these 248 pulse heights was the model representation of the peaks shown in Figure 1.

### 2.2. Model for Fluorescence Response of Labeled Beads Transiting the Laser Beam of a Flow Cytometer

The model of the flow cytometer response consists of a part that describes the beads and a larger part that deals with the flow cytometer instrument. The bead part contains the number of labels on/in a bead, the absorption cross-section and the quantum yield of the labels, and the standard deviation associated with each of the bead properties. The instrument part of the model consists of the laser power and the beam properties, the numerical aperture (NA) of the light collection optics [6], the PMT quantum efficiency and its gain (electron multiplication), optical filter properties, the current-to-voltage converter (CVC) gain, the analog-to-digital converter (ADC) bin size and resolution, and the algorithm to extract the associated pulse height from the digital pulse representation of the bead transit through the flow cytometer sensing volume. Lastly, it was necessary to deal with the background signal from blank beads which contain no labels. It was assumed that the signal from blank beads is also present in the beads with labels since all have in common auto fluorescence, Raman scattering from the bead material and solvent, and stray light from elastic scattering of the laser light. It is hard to characterize the background signal because the Raman scattering cross-section and autofluorescence quantum yield are not known. The stray light contribution depends on the instrument configuration. In the current model, the background signal was characterized by a phenomenological parameter called the effective number of label molecules (and its standard deviation), which reproduced the observed signal from the blank beads (bead B). A list of parameters used in the model is given in Table 3. Column 1 in Table 3 gives the parameter name, column 2 gives the nominal value, column 3 gives the place where the parameter is used, and column 4 gives a short description.

The FC response of the labeled beads was calculated in the model by adjusting the number and SD of the label molecules per bead until the response calculated with the model reproduced the experimental results. Since the background signal and the detection fluctuations were included in the model, the final value of the number of labels and its SD constituted the full characterization of the bead.

During the transit of a labeled bead through the laser beam of a flow cytometer, a stream of fluorescence photons was produced. Each of the photons originated from a randomly occurring absorption–emission event, and the time of occurrence of the emission event was recorded in an array. After the bead passed the laser beam, the array of photon emission times was analyzed to produce a histogram with the horizontal axis containing bins of time intervals starting from the time the bead entered the laser beam. The vertical axis contains the number of photons whose emission times fell within a given bin. The histogram was further processed by converting the number of photons in a bin to an electron current, converting the current to a voltage, and digitizing the voltage pulse to obtain a time series of digital numbers which represent the voltage pulse. Section 2.2.1 describes the absorption of a photon from the laser beam by dye molecules in the bead. Section 2.2.2 describes the radiative relaxation of the excited state of the label and the production of the photon stream. Section 2.2.3 describes the photon optical path from the bead through the detector aperture, passage through filters, and arrival at the PMT. Section 2.2.4 describes the photoelectron conversion at PMT cathode, and Section 2.2.5 describes the remaining steps in the generation of the voltage pulse shown in Figure 2.

#### 2.2.1. Calculation of the Total Optical Absorption Rate

The optical absorption rate by a single dye molecule depends on the intensity of the incident electric field and the transition dipole matrix element between the two states involved in the transition. The product of the energy difference between the two states and the optical absorption rate yields the total energy absorbed by a single molecule per unit time (energy/s). A commonly measured property of a single dye molecule is the optical absorption cross-section defined as the energy absorbed by the dye molecule per unit time (energy/s) divided by the total incident energy flux (energy/(s·cm^2^). Therefore, the energy absorbed per second by a single dye molecule can be written as the product of the optical absorption cross-section and the incident energy flux. A concise description of the process can be found in [7].

The light energy absorbed by the entire bead is the sum of light energy absorbed by the individual labels bound to the bead. The dots on the colored spheres in Figure 3 represent labels on the surface or inside the bead. Figure 3 shows seven consecutive positions of a bead traversing the laser beam. The dots are the result of the stochastic labeling model (assuming a random distribution throughout the bead volume). Each dot has a specific *x*-, *y*-, *z*-coordinate. The laser beam travels along the *z*-direction (out of the figure). The laser beam has a flat-top shape with intensity variation in the *x*–*y* plane and is polarized in the *y*-direction (Appendix A). The detector axis coincides with the *x*-axis. The cell travels in the *y*-direction, and, as it traverses the laser beam, the labels are illuminated. The illumination is calculated by simply evaluating the laser beam intensity at the location of each label. The optical absorption rate (*abs*_rate_) for the entire bead was calculated following Equation (1).
(1)$$abs_{\mathrm{rate}}=\frac{P_{o}}{A}\sigma \sum_{i}\exp(-2{\left(\frac{x_{i}}{w_{x}}\right)}^{2n})\cdot \exp\left(-2{\left(\frac{y_{i}}{w_{y}}\right)}^{2}\right)\cdot Gr_{i}\cdot \frac{\lambda }{hc}.$$

The index *i* runs over all labels in the bead, *P_o_* is the total laser power (the reduction in laser power due to absorption by the labels was neglected), A is the area of the waist of the laser beam through which the bead passes, *w_x_* and *w_y_* are the half-widths of the flat-top laser beam waist in the *x*- and *y*-directions, respectively, *n* is a power which controls the drop-off of the beam intensity in the *x*-direction [8], *n* was set to 5 to give a reasonable beam shape, σ is the average optical absorption cross-section, *λ* is the laser wavelength, *h* is Planck’s constant, and *c* is the speed of light in water. The laser wavelength, *λ,* was chosen so that the factor λ/hc is close to the energy difference between the excited and ground states of the label. The factor λ/hc in Equation (1) converts the expression to a rate of photon absorption. The factor *Gr_i_* is the occupation number of the ground state (i.e., 1 if occupied and 0 if empty, which corresponds to the dye in the excited state). Absorption can take place only if the molecular ground state is occupied. To summarize Equation (1), *P_o_* is divided by the area of the laser beam waist, multiplied by the optical cross-section, the beam shape factor, and the ground state occupation number, and divided by the energy of the molecular transition to yield the photon absorption rate for the entire bead. Each absorption leaves the dye molecule in the excited state, which can relax back to the ground state either via the emission of a fluorescence photon or by transferring the energy to vibrational modes (internal conversion). The excited state can also undergo photochemical reaction resulting in a “dark” label [9]. The present version of the model does not treat photodegradation explicitly, nor does it consider stimulated emission from the excited state [10]. The model uses the measured quantum yield (QY) parameter to estimate the fraction of excited states which relax via emission of a fluorescence photon. Thus, the product of the optical absorption rate and the QY gives the rate at which fluorescence photons are produced during the transit of the cell through the focused laser beam. Section 2.2.2 provides a detailed model of the genesis of the stream of photons as the bead transits the laser beam. Keep in mind that the transit is described by the time dependence of the *y_i_* coordinate in Equation (1).

#### 2.2.2. Calculation of the Photon Emission Stream with the Stochastic Absorption–Emission Model

The stochastic measurement model was implemented by performing random absorption and emission events during the time the bead was illuminated by using the rule proposed by Gillespie [11,12]. There are two time periods to consider. The first is the time the bead spends traversing the laser beam. This time interval (about 10^−5^ s) is found by dividing the distance between two chosen spatial points which span the region of large laser intensity by the macroscopic linear velocity of the bead. The second time interval is the average random time interval (about 10^−10^ s) between any sequential pair of absorption/emission events occurring while the bead is transiting the laser beam. If the average interval between absorption/emission events is much smaller than the transit time through the laser beam, the rule given by Gillespie is valid (see Appendix B). In addition, the sum of all intervals between absorption/emission events should equal the bead transit time through the laser beam.

The circles centered on the origin in Figure 3 show a contour plot of a laser with a flat-top beam shape. Each of the seven bead images in Figure 3 was obtained by simply adding 10 × 10^−6^ m to the *y*-coordinate of the bead originally defined relative to the same origin as the center of the laser beam.

The total illumination was calculated by adding the illuminations of all labels in the bead according to Equation (1). The calculation of the photon stream was started with the bead’s *y*-coordinate at −30 × 10^−6^ m relative to the beam center (the intersection of the laser propagation direction and the detector center line defines the origin of the coordinate system for the bead and the laser beam). The absorption rate was calculated at this initial point according to Equation (1), and the time, *tau*, for the occurrence of the next absorption or emission event was determined by choosing a random number r_1_ in the range [0, 1] and using the Gillespie rule Equation (2).
(2)$$tau=\frac{1}{decay_{\mathrm{rate}}+abs_{\mathrm{rate}}}\log\left(\frac{1}{r_{1}}\right).$$

The total decay rate of the excited state, *decay_rate_*, was calculated using Equation (3) as the product of the inverse of the measured lifetime, τ, multiplied by the number of labels in the excited state.
(3)$$decay_{rate}=\frac{1}{\tau }\cdot \underset{i}{\sum }\left(1-Gr_{i}\right).$$

Note that *Gr_i_* = 0 for labels in the excited state. Each of the excited labels decayed to the ground state with a rate that is equal to the inverse of the lifetime, which, in the case of fluorescein molecules, is about 4.3 ns. Next, the reaction was chosen by generating a random number r_2_ and comparing it to the ratio $abs_{rate}/\left(decay_{rate}+abs_{rate}\right)$. If r_2_ was less than the ratio, absorption was chosen and Gr_i_ was set to 0 for a randomly chosen label with Gr_i_ = 1. If r_2_ was greater than the ratio, decay was chosen and Gr_i_ was set to 1 for a randomly chosen label with *Gr_i_* = 0. It is important to keep track of the occupation numbers of labels in the ground state and the excited state since these occupation numbers are used to calculate absorption and decay rates. The absorption and decay were treated as independent events. For a single label, absorption is always followed by decay. For a collection of many labels, the absorption by one label can be followed by absorption by a second label before decay of the first label takes place. To make the code faster, only a choice between absorption and decay was implemented at each step of the transit. The other decay pathways, such as internal conversion, transition to the triplet state, and the initial step of a photochemical reaction, were not considered explicitly in this model. The nonradiative decay pathways were taken into account later by choosing a random number, r_3_, for each photon emission time and discarding the photon if r_3_ was greater than 0.95, the value of the quantum yield (QY). The typical distribution of absorption rates, decay rates, and intervals between sequential events is shown in Figure 4. Next, the location of the bead was advanced by *v_o_ tau* where *v_o_* is the linear velocity of the bead (assumed to be 80 cm/s in the simulation), and the time was advanced to *t* + *tau*. If the event was a radiative decay, the time of the decay was stored in an array named *photont*. The whole process was repeated until the bead reached +30 × 10^−6^ m. The array *photont* contained the times of all radiative decays of the excited states during the transit of the bead through the laser beam. The simulation was checked for self-consistency by requiring that the simulated absorption rate tracks the beam profile, that the simulated bead moved with the expected macroscopic velocity, and that the number of absorptions and the number of decays differed by no more than 1 (the absorption or emission which occurred when the bead arrived at +30 × 10^−6^ m stopped the recording of further absorption or emission events). The *photont* array was used to estimate the number of fluorescence photons arriving at the cathode of the PMT detector during the transit time of the bead, as discussed in Section 2.2.3.

#### 2.2.3. Optical Path of the Photon Stream to the Photomultiplier (PMT) Detector

The numerical aperture of the detector optics was 1.2, giving a cone angle of 64° or 3.53 steradians relative to the location of the bead at the center of the laser beam. For every element in the photon emission time sequence array, *photont*, a random number was chosen and, if it was less than 3.53/4*π* = 0.281, where 4π is equal to the steradians subtended by a sphere, the photon was included in a new time sequence containing photons destined for the PMT. Otherwise, the photons were discarded.

No attempt was made to include details of the photon path through dichroic mirrors and other optical elements. The fluorescence photons are distributed over a range of wavelengths (fluorescence emission spectrum), while dichroic mirrors have a specified wavelength range over which they transmit/reflect incident photons. Therefore, there is a chance that the incident photon will not be transmitted/reflected through the dichroic mirror to the photodetector assigned to that fluorescence channel. The photon path between the bead and the PMT was on the order of several centimeters, but the time taken to transit this path was not added to the arrival time of the photons at the photocathode.

#### 2.2.4. Photomultiplier to Current-to-Voltage Converter (CVC)

The photons arriving at the photocathode of the PMT produced electrons via the process of photoconversion. The photocathode quantum efficiency was set to 0.8 and the electron multiplication was set to 1.0 × 10^5^. A thorough description of the photomultiplier operation was given in [13]. A random number r_1_, between 0 and 1, was chosen for every photon arriving at the detector photocathode, and, if r_1_ was less than 0.8, a single electron was added to the electron stream leaving the photocathode. If r_1_ was greater than 0.8, the photon did not produce an electron and was discarded. After the electron multiplication in the PMT, the treatment of the electron stream was deterministic, and the electronic noise of various circuits was not considered in this version of the model. The sequential times of the electron stream leaving the photocathode were binned into 2 µs wide bins. The average time interval between two electrons (same as photon decay times) was 0.85 ns, as shown in Figure 4. The PMT is designed to handle such a stream of electrons as a current. The total number electron emission times in every 2 µs bin was multiplied by the PMT electron multiplication factor and divided by 2 µs (time interval of the bin) to give the final electron current (electrons per second) leaving the PMT. Lastly, the electron current was divided by 6.241 × 10^18^ electrons/coulomb to convert the electron stream to current in units of amperes (coulombs per second). The current was processed by a current-to-voltage converter (CVC) represented by an operational amplifier with a feedback resistor, R_f_, equal to 0.125 × 10^6^ Ω. Multiplying the current in each bin by the value of the feedback resistor gave the corresponding voltage associated with that bin. The measured voltage drop across the resistor is proportional to the difference in the energy between electrons entering one resistor terminal and electrons leaving the ground resistor terminal, which serves as a reference point. Large electron streams at the ground terminal may produce small voltage “baseline” shifts which are ignored in the current version of the model. The model also ignored electronic signal filters which are often used to reduce noise by limiting the frequency response of the amplifier. The time sequence of all voltage values resulted in a voltage pulse which represented the emitted photo stream from a bead traversing the FC laser beam.

#### 2.2.5. From CVC to a Digital Representation of Fluorescence Intensity (FI)

The voltage pulse was digitized using an analog-to-digital converter (ADC) which sampled the voltage pulse in each of the 2 µs time bins and put out a sequence of binary digits, FI, representing the voltage. The relationship between the voltage and the FI was FI = voltage × 2^16^/5, where 2^16^/5 means that the ADC was able to split 5 V into 2^16^ distinct digits. This resulted in a dynamic range of about 10^4^. The digitized FI values were fitted to a function given by Equation (4)
(4)$$FI=H*\exp\left(-\frac{{\left(time-mean\right)}^{2}}{width^{2}}\right).$$

The parameter H was the voltage pulse height and was the final representation of the pulse of fluorescence photons associated with the bead transit through the laser. The sequence of digitized values of a voltage pulse is shown by the solid circles in Figure 2, and the solid curve shows the best fit to the function given in Equation (4). The flow cytometer (FC) analysis consists of passing thousands of beads through the FC, determining the voltage pulse height for each bead, and forming a histogram of the heights of the pulses. Figure 5 shows a histogram for a sample of 1028 beads, with each bead producing a voltage pulse similar to that shown in Figure 2. The blue circles in Figure 5 were generated with model parameters: N_lb_ = 364//19, N_bg_ = 3.6/1.2, QY = 0.95, detector solid angle = 3.53 steradian, QE = 0.8, PMT electron gain G = 1.0 × 10^5^, current-to-voltage converter (CVC) gain = 0.125 × 10^6^, and analog-to-digital converter (ADC) resolution = 5 V into 2^16^ distinct binary digits. The orange circles in Figure 5 were generated with a constant value N_lb_ = 364, and all other parameters the same as used to generate the blue circles.

The blue circles in Figure 5 are a model representation of the measured MFI peak, M, shown in Figure 1. The model calculates the response of one bead population at a time, and the histogram in Figure 5 uses linear bin space. The histogram in Figure 1 was formed using the log of the pulse heights given in the fcs file, and 512 histogram bins were distributed uniformly in log space. The differences in histogram properties were considered when comparing the measured average height obtained from the fcs file and the model calculation.

### 2.3. Results of the Stochastic Model Calculations

The stochastic model was used to investigate the relationship between variations in parameter values and the resulting CV% of the FI. The CV% was obtained from a histogram similar to that shown in Figure 5 as a function of 1024 (or fewer) calculations of bead transits through the flow cytometer sensing volume. Each set of 1024 calculations was performed with the same combination of fixed/fluctuating model parameters. The calculated histogram (such as that shown in Figure 5) provides an estimate of the mean fluorescence intensity (MFI) and the standard deviation (SD) of the FI values in the histogram. Together, the two statistics yield the CV%, defined as 100 × *SD/MFI*. The next section deals with the contribution of the fluorescence detection process and fluctuations in the bead properties to the total measured CV%.

#### 2.3.1. Dependence of Variance on MFI

The model of the flow cytometer response consists of a part that deals with the flow cytometer instrument and a smaller part that describes the beads. The bead part contains the number of labels on a bead, the absorption cross-section and the quantum yield of the labels, and the standard deviation associated with each of the bead properties. The instrument part of the model was discussed above. Lastly, it is necessary to deal with the background signal from beads, which is estimated by measuring the response of blank beads without labels. It is assumed that the background signal from blank beads is also present in the beads with labels since all have in common auto fluorescence, Raman scattering from the bead material and solvent, and stray light from elastic scattering of the laser light. It is hard to characterize the background signal because the Raman scattering cross-section and autofluorescence quantum yield are not known. The stray light contribution depends on the instrument construction, and there may additionally be small fluctuations in the bead path through the laser beam resulting in small fluctuations in the fluorescence signal. In the current model, the background signal was characterized by a phenomenological parameter called the effective number of bead label molecules (and its standard deviation), which reproduced the observed background signal from blank beads.

The response of each labeled bead was calculated by adjusting values of the number of labels on each bead and the SD until the response calculated with the model reproduced the measured response shown in Table 4. The blank bead, B, was characterized by an effective number of labels and the corresponding SD. The resulting values of the number of labels and their standard deviations constituted the full characterization of the bead and are presented in columns 8 and 9 in Table 4 under the columns labeled *Bead properties* for the case of a flat-top laser profile and a flow rate of 100 μL/min. The model calculations in Table 4 include three varying quantities: the variation in the number of labels in each bead, the variation in background signal, and detection variation. The fluctuations in the bead properties were modeled by assigning random values for each bead transit through the laser beam. The detection fluctuations were built into the model and were not changed during the calculation.

The bead properties in column 8 and 9, the background signal, and the detection statistics lead to the values of calculated response shown in columns 5, 6, and 7. The data in columns 2 and 4 were reproduced from Table 2 and were used to make the blue circles in Figure 6. The values in columns 5 and 7 were used to make the orange circles in Figure 6. The data in Figure 6 were fitted with a quadratic function using Matlab program *lsqcurvefit* with a constraint that all parameters be positive. The best-fit polynomial is shown in the inset of Figure 6. The quadratic function was rationalized by a heuristic model as the proper description of the dependence of the variance on MFI, and the coefficients of the various powers were interpreted in terms of instrument and bead properties [3]. The coefficient of the quadratic term was interpreted as the square of the CV of the label number in the beads. The CV was a constant for all beads because the heuristic model assumed that the bead SD was proportional to the number of labels. The value of bead CV obtained from the quadratic fit was 0.05, which is close to the average value 0.052 of the calculated CV%/100 in column 6 of Table 4 (for L, M, and H beads). However, there is a trend in the bead CV% values indicating a slight decrease with increasing MFI. Since detection and background fluctuations were accounted for in the model, the slight decrease is most likely a bead property. The coefficient of the linear term in the quadratic fit, 5.75, was interpreted in the heuristic model as the inverse of the ratio of photoelectrons per unit of light intensity. The stochastic model prediction of the value of parameter c_1_ is 8.0 and is discussed in Appendix C. The value of c_o_ is effectively undetermined by the fit to the responses of the three beads L, M, and H. This suggests that a good estimate of the background parameter needs data from more beads with numbers of labels less than that of bead L. Overall, the observed behavior of the data shown in Figure 6 is consistent with the model calculations, giving credence to the model’s validity since the model has many parts which must work together to yield the prediction of the bead response in a flow cytometer. The measured and calculated values of bead CV% in Table 4 are close, suggesting that most of the measured CV% is due to fluctuations in bead label number. This is supported by the result that CV% = 0.2 for model calculations with the label numbers fixed. To increase the calculated CV% to 5.0, it is necessary to include fluctuations of the label number. Therefore, the detection and background variabilities are small when combined with the variability due to label number. The calculation of the properties of the H bead in Table 4 was restricted to small bead samples (less than 64) due to the long calculation time (about 10 min) required for each passage of the H bead through the laser.

#### 2.3.2. Interpretation of the Bead CV%

The stochastic model was used to calculate the MFI and the SD for various label conditions. The results are shown in Figure 7 for bead M, where each bar in Figure 7a gives the MFI for 128 independent calculations for the same label condition, as indicated at the bottom of the bar. For example, the first bar gives the result of 128 calculations for the case where N_lb_ = 364, SD = 19, and N_bg_ = 4, SD = 2. The second and third bars correspond to calculations with some of the SD values set to zero. The last bar gives the result for the case where N_lb_ = 364 while N_bg_ and all the SD values are set to zero. The first bar represents the realistic case where all label values fluctuate, and the last bar represents the ideal case where all label numbers are constant and only detection statistics contribute to the SD of MFI, resulting in CV% = 0.45. Adding fluctuation to N_bg_ increases the resulting CV% of MFI to 0.57. The results shown in Figure 7b suggest that, while N_bg_ has a measurable effect on MFI, its contribution to CV% is minimal. The dominant contributions to measured values of both MFI and SD come from the label number and its fluctuation. This observation suggests that the measured values of MFI and SD, shown in Table 2, can be converted to values associated with the number of labels on the beads by using procedure to subtract the contribution of background from the measured values of MFI and SD. The proposed procedure is discussed below.

It is highly likely that the background and labeled bead signals, compiled in Table 1 and Table 2, originate from two independent sources. In that case, the measured background values of MFI and CV% can be used to obtain the MFI and CV% values for the bead fluorescence alone. Equation (5) shows the explicit relation for the L bead at a flow rate of 100 µL/min. The index *L* refers to the bead name and *m* refers to the fact that it is the measured value given in columns of Table 1 and Table 2. The symbols *MFI_L_, SD_L_*, and *CV%_L_* refer to corrected bead values which have no contribution from the background and are given by Equation (5).
(5)$$MFI_{L}=MFI_{L,m}-MFI_{B,m}.$$
$$SD_{L}^{2}=SD_{L,m}^{2}-SD_{B,m}^{2}.$$

The values of *CV*%*_L_* were obtained by substituting the expressions given in Equation (5) into the definition of CV% given by Equation (6).
(6)$$CV{\%}_{L}=\frac{SD_{L}}{MFI_{L}}\times 100.$$

The calculated values, *MFI_L_* and *CV*%_*L*_, are shown in row “L” of Table 5 in columns “MFI” and “CV%”, respectively. Table 5 also contains the corrected values of MFI and CV% for the M and H beads. The values in Table 5 were compared to the results obtained from the stochastic model with the background set to zero. For example, 256 calculations for the flat-top laser and bead M with N_lb_ = 364/19 and N_Bkg_ = 0/0 gave MFI = 8536 and CV% = 5.2. The calculated values are in good agreement with the values MFI = 8520 and CV% = 5.2 shown in Table 5 for bead M and a flow rate of 100 µL/min.

The good agreement between analysis using Equation (5) and the results of the model calculations without background suggest that Equation (5) can be a useful method to analyze measured data. The detection contribution to the CV% in Equation (5) was neglected as suggested by model calculations.

#### 2.3.3. Interpretation of CV% Measured for Samples of Lymphocytes

The stochastic model was applied to the analysis of measurements of labeled samples of peripheral blood mononuclear cells (PBMC), obtained from two commercial sources. PBMCs are slightly larger than the beads with an average lymphocyte radius of 7.2 µm compared to the bead radius of 5.2 µm. The labels were located on the surface of the lymphocytes rather than dispersed throughout the bead. The scattering properties of cells and beads are similar, with the larger radius of the cells offset by a smaller difference in the index of refraction of the cell material relative to the aqueous medium surrounding the cells. Therefore, the results of the bead modeling can be applied to the analysis of the PBMC data.

The FC analysis resulted in approximately 5000–50,000 events. The signals in the R-PE-H fluorescence channel were gated with a combination of signals in the FSC-A and SSC-A channels. The symbols A and H in the above channel designation refer to the area and height of the photon pulse produced by the transit of the lymphocyte through the focused laser beam. The blue trace in Figure 8 shows the histogram of R-PE-H signals gated with the cell scattering events inside the polygon in the FSC–SSC scatter plot. The peak at FI digital number of about 200 is due to autofluorescence while the peak at digital number of about 20,000 is due to R-PE fluorescence.

The orange trace in Figure 8 is the autofluorescence peak obtained from the unstained Cyto-Trol sample with R-PE signal gated with the polygon around the scattering events in the FSC–SSC scatter plot. A similar graph was also obtained for the Vericell sample. The background correction for the blue peak at 20,000 in Figure 8 was performed using Equation (5) with the weighted means and standard deviations obtained from the R-PE and background peaks. The results are given in Table 6.

The mean (MFI) and the corresponding variance (Var) were obtained from a weighted average of the entries in fluorescence histograms discussed above.
(7)$$MFI=\frac{\sum_{i}\;bins_{i}\;\cdot \;counts_{i}}{\sum_{i}\;counts_{i}}.$$
(8)$$SD=\sqrt{\frac{\sum_{i}{\left(bins_{i}-MFI\right)}^{2}\cdot counts_{i}}{\sum_{i}counts_{i}-1}}\equiv \sqrt{Var}.$$

The SD, defined as the square root of the variance, was used to obtain an estimate of the uncertainty in MFI and in SD itself, as shown in Equations (9) and (10). The value of *events* in Equation (9) is given by the sum of all counts in the peak, $events=\sum_{i}count_{i}$. Assume that each of the counts in the bins is a normally distributed random variable. Then, the uncertainty in MFI is given by Equation (9) and the uncertainty in SD is given by Equation (10) [14]. Once the uncertainties in MFI and SD are evaluated, the uncertainty in CV% is given by the standard uncertainty propagation formula reproduced as Equation (11).
(9)$$sig_{MFI}=\frac{SD}{\sqrt{events}}.$$
(10)$$sig_{SD}=\frac{SD}{\sqrt{2\left(events-1\right)}}.$$
(11)$$sig_{CV\%}=CV\%\cdot \sqrt{{\left(\frac{sig_{SD}}{SD}\right)}^{2}+{\left(\frac{sig_{MFI}}{MFI}\right)}^{2}.}$$

The results in Table 6 show that the corrected values of MFI for the two samples of PBMC were 19,600 ± 350 for Vericell and 18,300 ± 254 for Cyto-Trol. These values of MFI are within two standard deviations of each other. The value of CV% for Cyto-Trol was 57.8 ± 2.2, which is significantly higher than 38.7 ± 2.4 for Vericell. This suggests that the MFI and CV% contain different information about the samples. According to the bead analysis, the CV% values are most likely associated with the CV% of the number of labeled CD19 receptors on the surface of B cells. The difference in the CV% between the PBMC samples from the two commercial sources is a quantitative result and most likely arose from differences in cell processing.

The model could be used to calculate the response of B cells labeled with R-PE conjugated to monoclonal CD19 antibodies. The values of the absorption cross-section and the quantum yield would have to be changed to values appropriate for the R-PE label. The radius of the cell would have to replace the radius of the bead, and some of the flow cytometer parameters would have to be adjusted to yield the expected number of labeled CD19 antibodies on reference B cells, approximately 8000 antibodies bound per cell (ABCs) [15,16]. The modified model, with inclusion of electronic noise, can be used to estimate the limit of detection of R-PE label on B cells, which is expected to be on the order of 10 R-PE molecules [17].

## 3. Discussions

The stochastic model of a flow cytometer (FC) reproduced the expected results of the analysis of a multilevel bead set, with each member of the set possessing a progressively larger number of labels. The parameters describing the operation of the flow cytometer were chosen to conform to the specifications of the various components of a generic flow cytometer. The main conclusion from the work was that the observed CV% of the detected MFI of multilevel bead set was mainly due to the CV% of the number of fluorescent labels on each of the beads. The contributions to the CV% due to background and detection fluctuations were minor corrections. To the best of our knowledge, this is the first comprehensive study of the properties of CV% of MFI in flow cytometer measurements. Both the source and the detection of the fluorescence signal were treated in a systematic manner.

The measurement and analysis of B cells labeled with anti-CD19 monoclonal antibodies (mAbs) followed the results of bead measurements. The conclusion is that the CV% from labeled cells is mainly the result of variation in the number of labeled mAbs on the cell. The observed difference in the CV% from lyophilized PBMC B cells obtained from two commercial sources suggests differences in the processes used to prepare the samples for FC analysis. The possibility arises of using a PBMC standard for quantitating and interpreting the differences in the CV% observed in fresh patient PBMC samples. This is based on the expectation that the CV%, observed in the FC analysis of labeled cells, can provide information on the nature of the receptor variation in PBMC samples from diseased donors compared to samples from normal donors. There is also the enticing possibility that significant biological information can be gleaned from the measurement of MFI and CV% in cell assays. Recent theoretical developments suggest that the MFI and CV% may be fundamentally related in biological processes [18]. The explicit relationship, adapted to FC measurements, is shown in Equation (12) [19].
(12)$$\frac{MFI_{a+\Delta a}-MFI_{a}}{{\left(CV{\%}_{a}\times MFI_{a}\right)}^{2}}=b\times \Delta a.$$

The symbol Δ*a* stands for a change in some biological parameter which leads to a change in the value of MFI. The magnitude of the change is determined by the value of *b*. *MFI_a_* and *CV*%_*a*_ are measured values at some value of *a*, and *MFI*_*a*+Δ*a*_ is the measured value of MFI at a changed parameter value *a* + Δ*a*. The relationship in Equation (12) is very general and adapting it to a specific biological process will require care. The original application of Equation (12) considered the change in green fluorescing protein (GFP) fluorescence due to a change in the sequence of a peptide attached to the front of the GFP gene [19]. It was surmised that changes in the peptide sequence changed the replication rate of the GFP in the host yeast cell. It is likely that well-designed studies can be carried out to investigate the interaction of labeled mAbs and their receptors on lymphocyte surfaces [20].

In summary, the study suggests that labeled cell measurements can be analyzed using Equation (5), which neglects the contribution from detection fluctuation (for larger values of MFI). Flat-top lasers and advances in microfluidics and electronics have significantly reduced detection fluctuation. The stochastic model simulated a flow cytometer with two scattering channels and one fluorescence channel without the need for compensation correction. The stochastic model can be extended to multicolor flow cytometers by replicating the model for each of the additional channels, considering the complicated optical pathways and implementing compensation corrections. Compensation correction is a procedure for estimating the background signal in each fluorescence channel due to the presence of dye molecules associated with other fluorescence channels. It is expected that the CV% values in multicolor flow cytometer measurements can be quantitated and should become an important component of flow cytometer analysis. Lastly, the stochastic model can be used to clarify the interplay of noise, background, and signal when the fluorescence signal is low.

## 4. Materials and Methods

The Attune NxT flow cytometer (ThermoFisher Scientific, Waltham, MA, USA) was used to analyze a sample containing four latex bead populations, each with progressively larger loading of dye molecules. The beads were obtained as a special order from Thermo Fisher Scientific. The bead diameter was 5.2 µm. The fcs files were read using the *fca_readfcs* program [21] and analyzed using MATLAB (Natick, MA, USA). The horizontal axis, called BL1-H, shows the height of the fluorescence photon pulse detected by the BL1-H channel when a bead passes the 488 nm blue laser beam. The vertical axis shows the number of times a particular signal was observed during the FC analysis of the bead sample. The histogram was created by first using MATLAB function *histocounts* to get the edges of 512 bins of log-scaled BL1-H data, and then using the function *histogram* to bin the data using linear values of the edges. In practice, about 10,000 beads were included in the analysis.

The PBMC samples (Cyto-Trol, from Beckman Coulter Life Sciences, Indianapolis, IN, USA; Vericell, from BioLegend, San Diego, CA, USA)) were resuspended according to the prescribed protocol of the manufacturer and were labeled with a CD19-R-PE dye. The antibody to CD19 binds to the CD19 surface antigen, which is expressed mainly by B cells. The samples of PBMC were analyzed on a Attune NxT flow cytometer. The flow cytometer was run with the simplest possible configuration, including forward scattering channels (FSC-A, FSC-H), side scattering channels (SSC-A, SSC-H), and a fluorescence channel (R-PE-H) for detecting the R-PE fluorescence emission. Compensation correction was disabled.

## Figures and Tables

**Figure 1 ijms-22-08256-f001:**
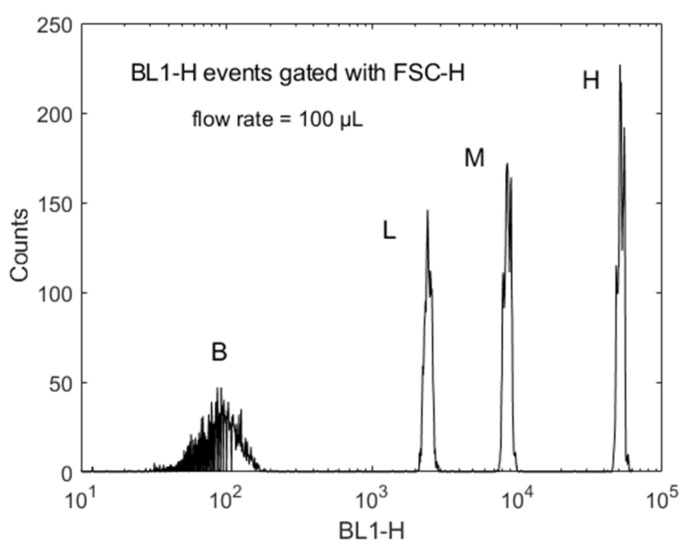
A histogram of the signals recorded by a flow cytometer for a sample consisting of a mixture of four multilevel beads B, L, M, and H. B stands for a blank bead, whereas L, M, and H stand for beads with low, medium, and high loading of label molecules. BL1-H is the digital representation of the height of the voltage pulses detected in the first fluorescence channel of the 488 nm laser. Counts represent the number of beads which gave a signal associated with the value on the BL1-H axis. The histogram has 512 bins along the BL1-H axis and a total of 10,000 events. The sample flow rate was 100 uL/min. Each fluorescence event was gated with a strong signal in the forward scattering channel, FSC-H. There was minimal electronic noise recorded on the histogram. The signals from all beads were large; thus, the gain was set low.

**Figure 2 ijms-22-08256-f002:**
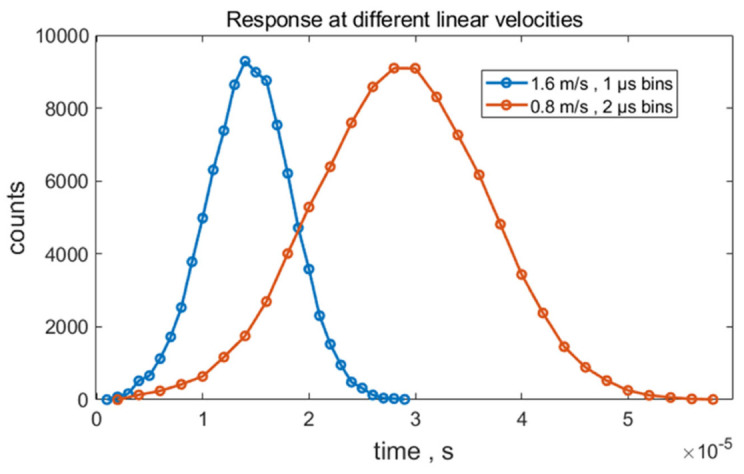
Model calculation of the time sequence of outputs from the analog-to-digital converter for a single transit of a bead through the laser beam. The pulse height is about 9100 digital units and represents one event on the histogram in Figure 1. The blue and orange dots are for beads with different linear velocities. The model automatically adjusts the time bin width to give the same pulse height at different linear flow velocities.

**Figure 3 ijms-22-08256-f003:**
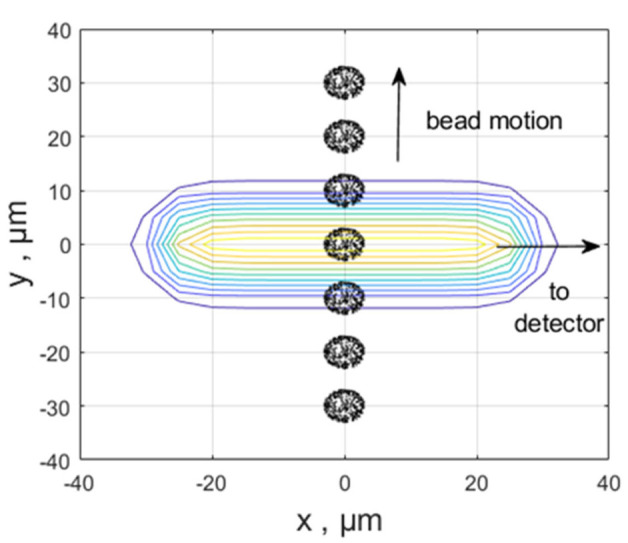
The black spheres show seven consecutive positions of a bead traversing the laser beam. The ellipsoids centered on the origin show a contour plot of a laser with a flat-top beam shape. Each of the bead images was obtained by simply adding or subtracting 10 × 10^−6^ m to/from the *y*-coordinate of the bead originally defined relative to the same origin as the center of the laser beam. The total illumination was calculated by adding the illuminations of each label on the bead. The illumination calculation was performed by inserting the array of bead coordinates (after modification of the *y*-coordinates) into the flat-top function describing the laser beam intensity.

**Figure 4 ijms-22-08256-f004:**
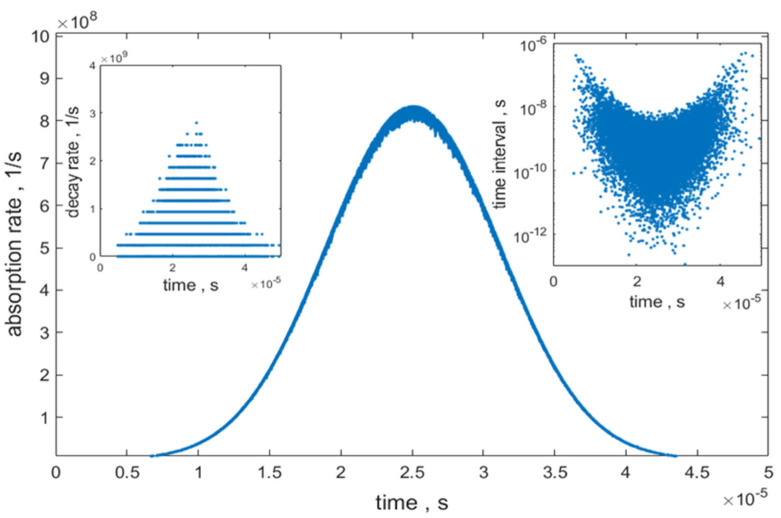
Results of calculated absorption and decay rates during the bead transit The absorption rates are displayed by the central trace while the decay rates are shown in the left inset. The decay rates are staggered due to different occupation numbers of excited states during the bead transit time of the laser beam. The time is started when the center of the bead is at −30 µm in Figure 3 and ends when the bead reaches +30 µm. The inset on the right shows the distribution of time intervals between any two sequential absorption/decay events.

**Figure 5 ijms-22-08256-f005:**
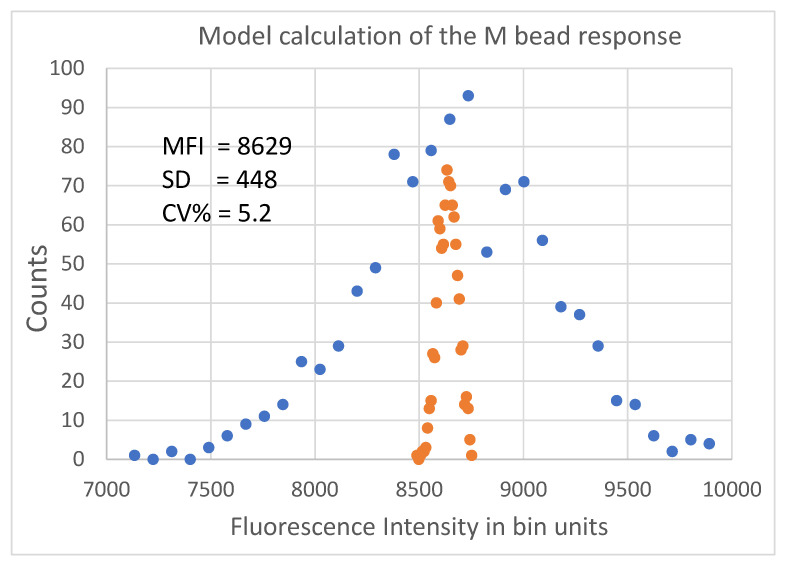
Histogram of the calculated heights of the digitized voltage pulses of 1024 M beads passing the laser beam. The horizontal axis plots the height of the pulse in bin unit and the vertical axis gives the number of times the height occurred. The bin values range between 1 and 65,536. Bead M occupies a small portion of the bin range and has an average height equal to about 8600 digital bin units. For the blue circles, the number of labels on the bead was 364 with a label SD of 19. For the orange circles, the number of labels on the bead was set to a constant value of 364. Clearly the variability of the number of labels on a bead is important.

**Figure 6 ijms-22-08256-f006:**
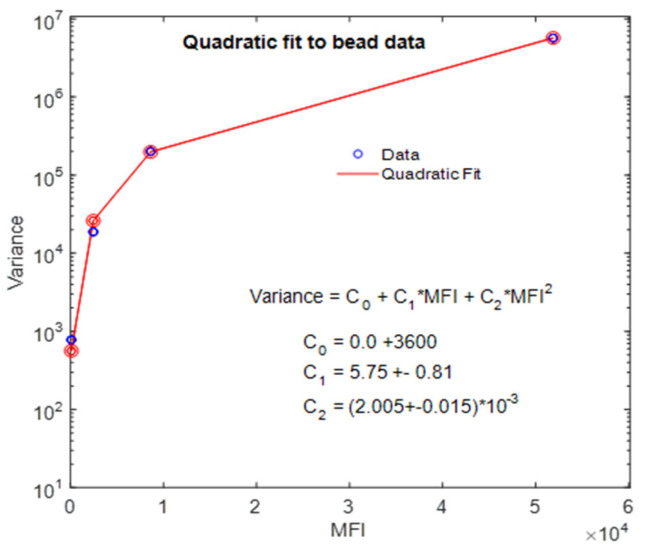
The blue circles show the measured values of the variance plotted against the corresponding values of MFI. The orange dots show the variance and MFI obtained from the model. In most cases, the orange circles overlay the blue circles. The red trace is the fit to a quadratic polynomial with the fitted function shown in the inset. The plot was made using data in Table 4.

**Figure 7 ijms-22-08256-f007:**
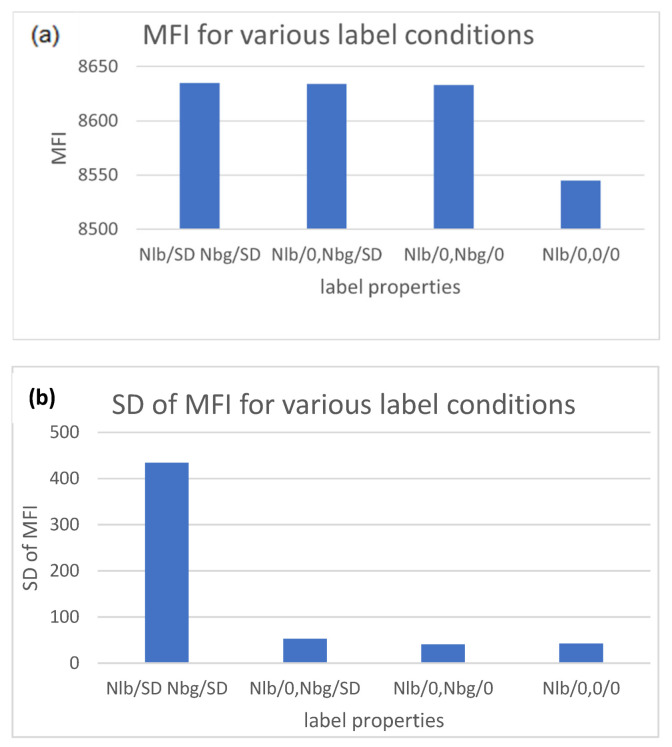
(**a**) Each bar gives the value of MFI from 128 independent calculations for bead M with label conditions specified at the bottom of the bar. A zero means that the specified value was set to 0 in the calculation. (**b**) Each bar gives the SD of the MFI value shown in (**a**) with the same label properties.

**Figure 8 ijms-22-08256-f008:**
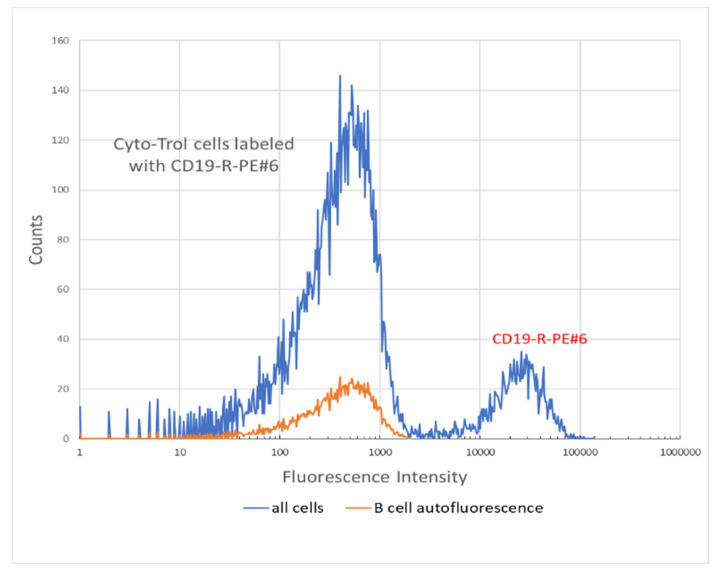
Histogram of fluorescence signal from a sample of PBMC labeled with CD19-R-PE #6. The orange trace is an estimate of the autofluorescence from B cells alone. The horizontal axis gives the fluorescence intensity in digital units.

**Table 1 ijms-22-08256-t001:** Laser with a Gaussian beam profile.

	100 µL/Min		200 µL/Min		500 µL/Min	
	MFI	CV%	MFI	CV%	MFI	CV%
B	133	36.4	130	36.4	134	36.2
L	1861	7.2	1877	7.1	1889	7.3
M	6521	5.9	6568	6	6588	6.1
H	38,882	5.4	39,373	5.5	39,461	5.6

**Table 2 ijms-22-08256-t002:** Laser with a flat-top beam profile.

	100 µL/Min		200 µL/Min		500 µL/Min	
	MFI	CV%	MFI	CV%	MFI	CV%
B	95	29.4	88	43.2	85	39.7
L	2449	5.6	2455	6.5	2452	6.7
M	8615	5.2	8628	5.3	8626	5.3
H	51,882	4.6	51,899	4.8	51,880	4.8

**Table 3 ijms-22-08256-t003:** Parameters used in the stochastic model of flow cytometer.

Name	Value	Location	Description
*P* _0_	0.04 W or 0.1 W	laser	Power of illuminating laser
*λ*	488 × 10^−9^ m	laser	Wavelength of illuminating laser
*w_x_ w_y_*	25 µm, 10 µm	laser	Half-widths of the beam waist of the flaser beam
*delx*	varies	bead/cell	Deviation of path of bead/cell through laser beam
*N_lb_*	varies	bead/cell	Number of labels on the surface of the bead/cell
*N_bk_*	varies	bead/cell	Number of effective background labels
*sig*	3.06 × 10^−20^ m^2^	bead/cell	Absorption cross-section of FITC label
*QY*	0.95, varies	bead/cell	Fluorescence quantum yield of FITC label
*τ*	4.3 × 10^−9^ s	bead/cell	Fluorescence decay lifetime of FITC label
*R_b_*	2.6 × 10^−6^ m	bead/cell	Radius of bead (diameter = 5.2 µm)
*v* _0_	0.80 m/s	bead/cell	Linear velocity of bead in sample stream
*ster*	3.53 steradian	detector	Solid angle of detector aperture
QE	0.8 varies	detector	Quantum efficiency of PMT photocathode
Gain	1.0 × 10^5^	detector	Photoelectron multiplication by PMT
*R_f_*	1.25 × 10^5^	detector	Gain of the current-to-voltage converter
Resolution	5/2^16^	detector	ADC amplitude resolution, 5 V into 2^16^ bins
Time bin	2 × 10^−6^ s	detector	ADC time bin width
c	2.26 × 10^8^ m/s	constant	Speed of light in water (m/s)
h	6.63 × 10^−34^ J·s	constant	Planck constant (J·s)

**Table 4 ijms-22-08256-t004:** Measured and calculated response. The number of labels on L, M, and H beads were adjusted to yield the calculated response shown in the table. The properties of the background bead, B, remained unchanged for the calculations of the other beads.

Name	Measured Values, 100 μL/Min			Calculated Response			Bead Properties		
Bead Name	MFI	CV%	Variance	MFI	CV%	Variance	Number of Labels	SD	CV%
1	2	3	4	5	6	7	8	9	10
B	95	29.4	780	92	28.4	729	3.6	1.2	33
L	2449	5.6	18,808	2451	5.8	19,881	100	5.4	5.9
M	8615	5.2	200,686	8606	5.1	200,704	364	19	5.2
H	51,882	4.6	5,695,726	51,709	4.6	5,657,804	2115	91	4.3

**Table 5 ijms-22-08256-t005:** Corrected values for flat-top laser beam profile.

	100 µL/min		200 µL/min		500 µL/min	
	MFI	CV%	MFI	CV%	MFI	CV%
L	2354	5.7	2367	6.5	2367	6.8
M	8520	5.2	8540	5.3	8541	5.3
H	51,787	4.6	51,811	4.8	51,814	4.8

**Table 6 ijms-22-08256-t006:** MFI and SD of Cyto-Trol^®^ and Vericell^®^ samples.

	Cyto-Trol^®^		Vericell^®^	
	Uncorrected	Corrected	Uncorrected	Corrected
MFI	18,400 ± 245	18,300 ± 254	19,700 ± 350	19,600 ± 350
SD	10,600 ± 175	10,600 ± 175	7600 ± 250	7600 ± 250
CV%	57.6 ± 2.2	57.8 ± 2.2	38.6 ± 2.4	38.7 ± 2.4

## Data Availability

Software used in the model calculations are available from Lili.Wang@nist.gov and Adolfas.Gaigalas@gmail.com.

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
