# Peer review of "Sources of Variability in the Response of Labeled Microspheres and B Cells during the Analysis by a Flow Cytometer"

_ijms, 2021, doi:10.3390/ijms22158256_

Round 1

Reviewer 1 Report

The overall impression is that this manuscript concerns the interesting topic of factors contributing to MFI variation in Flow Cytometry. It is written by a group with long standing and impressive previous experience in stochastic models (as deduced by their previous publications), who seem to have conducted a deep and thorough investigation of the subject. From the flow cytometrists perspective, I cannot find any major issues to argue against.

However, I have strong objections as to the structure of the present manuscript:

  1. The introduction does not seem to set the frame of the study successfully. Which is the basic question to be answered? (From my point of view, building a stochastic model is a means to investigate non biological variation of MFI; it is not the question per se). Why is it important to investigate it? Which is the current level of relevant knowledge? (the authors refer mainly to their own work)
  2. I have found the structure of the text not always having a logical flow. That is, although most information is there, it is not always presented in an organized way, especially in the initial sections: (a) Many paragraphs do not have any introductory sentence. As a result, I had to read further beyond to understand the basic subject of most of them. (b) Some combine many issues together (from referring to some of the variables in question to instrumentation issues and back to some results as, just to give an example, in line 146). (c)This results in many repetitions (lines 141-2 and 380-81 are an example of a copy/pasted sentence). (d) Moreover, it seems that reference to various issues is disproportionate (ie there is a detailed description of the histogram in 83-86, but there is no description of the flow cytometer). (e) My overall impression is that the manuscript is an assortment of paragraphs written for separate projects. Though, in general, they are individually detailed and to the point, their combination is not that successful.

 I would suggest that the authors initially describe the main events of generation and detection of fluorescence all together and link them to the variables they use (preferably by including a comprehensive figure). Then, describe how they calculate their variability designating a separate paragraph to each one. I would believe thar a more brief, concise, and organized rewriting of the text is needed.

  1. There is a list of assumptions made throughout the study. They are scattered all over the text (as for example in lines 58, 188, 197 and 214). It could be convenient to group them and provide a list of variables that were not taken into account and comment on that.
  2. Most importantly there is a lack of adequate references: (a) Although the selection of most variables is based on classic Fluorescence and Flow cytometry knowledge, it should be supported by classic references (ie lines 47-73 describe multiple variables without a single reference). (b) Given the multitude of classic publications, the use of a commercial company’s website as a reference for Flow cytometry strongly compromises the validity of any argument (lines 74-75, reference 4). (c) When referring to textbooks (ref 7), it would be useful to be more particular (pages?, chapter?). (d) The authors refer to their own publications, without stating that. This gives the wrong impression that they use studies originating from multiple centers. (e) Comparison with similar studies from different centers would be welcome (otherwise the uniqueness of the study should be stated)  
  3. A list of Abbreviations would be welcome.

Author Response

Cover letter to Rev 1

We are grateful for the very constructive criticism provided by reviewer 1. In response, I changed reference 4 from a commercial pamphlet to “Flow Cytometry-A Basic Introduction” (written by M. Ormerod and hosted as an e-book on De Novo site).  De Novo is a commercial site for flow cytometer data analysis software and is very well known and respected. Chapter 2 of the e-book describes how a flow cytometer works. In addition, Chapter 4 and higher chapters deal with the myriad of applications of flow cytometry. Although the e-book was published in 2008, it is still an excellent overview of flow cytometry measurements. I am not aware of work on quantifying CV%. Furthermore, there are no standards for CV%. In my opinion, the best approach was to develop a model for flow cytometer measurements as a basis for understanding CV%. The manuscript was written with a very modest and narrow objective- understand CV% in flow cytometry measurements.

The flow cytometer measurement is a very complex process with many steps. Some of the steps have little in common and are treated by different disciplines. For example, the production of the photon stream (photo chemistry) and the conversion of current to voltage (electronics). We tried to minimize the appearance of disjoint nature of the process by a logical transition from one step to the next.

This work suggests that the CV% of MFI from measurements on cells provides a reliable measure of the heterogeneity of the number of bound labeled antibodies on the cells.

Included a list of Abbreviations extending the list given in Table 3.

Abbreviations (not included in Table 3)

FC           flow cytometer

FI            fluorescence intensity, height of a fluorescence pulse, volts or digital unit

MFI        mean fluorescence intensity

SD          standard deviation of the fluorescence signals

CV          coefficient of variation, SD divided by MFI.

CV%      the product of CV and 100

PMT       photomultiplier tube

CVC       current to voltage converter circuit

ADC      analog to digital converter circuit

Gri           ground state occupation index of the ith label. 1 if occupied, 0 if not occupied

X,Y,Z     names of the axis used to describe the geometry of the flow, laser propagation, and detector

x,y,z,      specific values of X,Y,Z coordinates

L,M,H    names of three beads, with low, medium, and high label loadings

PBMC    peripheral blood mononuclear cells

mAb      monoclonal antibodies

FITC       fluorescein isothiocyanate, derivative of fluorescein, a fluorescent molecule

R-PE      R-phycoerythrin, fluorescent protein isolated from red algae

CD19     receptor found on the surface of B cells

To the best of our knowledge, this is the first comprehensive study of the properties of CV% of MFI in flow cytometer measurements. Both the source and detection of the fluorescence signal were treated in a systematic manner.

Reviewer 2 Report

This manuscript describes the development of a stochastic model for the assessment of the nature of the observed coefficient of variation (CV%) of the mean fluorescence intensity from a population of labeled microspheres. Overall, the manuscript is well-written and easy to follow. The data and statistic analysis are solid and are presented in a logic manner. The reviewer supports the publication to the journal IJMS.

A minor issue: Figure 2 seems to be cut off on the bottom and will need to be corrected.

Author Response

Figure 2 was fixed